

# Coral responses to a repeat bleaching event in Mayotte in 2010

David O. Obura[1], Lionel Bigot[2] and Francesca Benzoni[3]

[1] CORDIO East Africa, Mombasa, Kenya
[2] UMR Entropie, Laboratoire d'écologie marine, Université de la Reunion, Saint Denis, Reunion
[3] Department of Biotechnology and Biosciences, University of Milano—Bicocca, Milan, Italy

## ABSTRACT

**Background**. High sea surface temperatures resulted in widespread coral bleaching and mortality in Mayotte Island (northern Mozambique channel, Indian Ocean: 12.1°S, 45.1°E) in April–June 2010.

**Methods**. Twenty three representative coral genera were sampled quantitatively for size class distributions during the peak of the bleaching event to measure its impact.

**Results**. Fifty two percent of coral area was impacted, comprising 19.3% pale, 10.7% bleached, 4.8% partially dead and 17.5% recently dead. *Acropora*, the dominant genus, was the second most susceptible to bleaching (22%, pale and bleached) and mortality (32%, partially dead and dead), only exceeded by *Pocillopora* (32% and 47%, respectively). The majority of genera showed intermediate responses, and the least response was shown by *Acanthastrea* and *Leptastrea* (6% pale and bleached). A linear increase in bleaching susceptibility was found from small colonies (<2.5 cm, 83% unaffected) to large ones (>80 cm, 33% unaffected), across all genera surveyed. Maximum mortality in 2010 was estimated at 32% of coral area or biomass, compared to half that (16%), by colony abundance.

**Discussion**. Mayotte reefs have displayed a high level of resilience to bleaching events in 1983, 1998 and the 2010 event reported here, and experienced a further bleaching event in 2016. However, prospects for continued resilience are uncertain as multiple threats are increasing: the rate of warming experienced (0.1 °C per decade) is some two to three times less than projected warming in coming decades, the interval between severe bleaching events has declined from 16 to 6 years, and evidence of chronic mortality from local human impacts is increasing. The study produced four recommendations for reducing bias when monitoring and assessing coral bleaching: coral colony size should be measured, unaffected colonies should be included in counts, quadrats or belt transects should be used and weighting coefficients in the calculation of indices should be used with caution.

Corresponding author
David O. Obura,
dobura@cordioea.net

# INTRODUCTION

Coral bleaching is an increasingly common phenomenon on tropical coral reefs as background warming occurs and inter-annual modes of climate variability intensify (*McPhaden, Zebiak & Glantz, 2004*; *Hoegh-Guldberg et al., 2007*). Repeated major coral

bleaching events are now standard occurrence in most world regions (*Donner, Rickbeil & Heron, 2017*; *Hughes et al., 2018*), including the Caribbean (*Eakin et al., 2010*; *Jackson et al., 2014*), the Pacific (*Chin et al., 2011*), and the western Indian Ocean (*McClanahan et al., 2014*; *Obura et al., 2017*), and recently in concurrent years on the Great Barrier Reef (*Hughes et al., 2017b*). The Western Indian Ocean (WIO) has suffered repeated bleaching events, with the most extreme being in 1998 (*Wilkinson et al., 1999*; *Goreau et al., 2000*), but with smaller events before, most notably in 1983 (*Faure et al., 1984*) and since, in 2005 (*McClanahan et al., 2005*), 2007 (D Obura, 2007, unpublished data), 2010 (*Eriksson, Wickel & Jamon, 2012*) and 2016 (*Nicet et al., 2016*; *Obura et al., 2017*). The increasing frequency of major bleaching events (*Hughes et al., 2018*) is calling into question the long term survival of coral reef ecosystems, with most world regions predicted to experience severe bleaching conditions on an annual basis within the next 40–80 years (*Van Hooidonk et al., 2016*).

How well coral reefs will cope with these conditions is a major question in current research (*Hughes et al., 2017a*). The WIO was among the worst affected regions during the 1998 bleaching event. In that year three phenomena coincided—1998 was an unusually hot year globally, and strong positive phases of the El Niño Southern Oscillation (ENSO) and Indian Ocean Dipole (IOC) occurred in phase (*Saji et al., 1999*; *McPhaden, Zebiak & Glantz, 2004*). As a consequence, coral mortality at different reef sites varied between 50–80%, and accounted for a loss of 16% of healthy reefs (reviewed in *Wilkinson, 2000*). The Northern Mozambique Channel, one of the least known regions of the WIO, had variable levels of coral bleaching reported in the 1998 event (*Souter, Obura & Linden, 2000*; *Wilkinson, 2000*; *Obura et al., 2018*). While it is a center of diversity and accumulation for coral species (*Obura, 2012*; *Obura, 2016*), it may show some characteristics of being a climate refuge because of its low rate of temperature rise (*McClanahan et al., 2007a*; *McClanahan et al., 2007b*; *McClanahan et al., 2014*). However, larger scale studies present it as among the earliest parts of the WIO to face severe thermal stress conditions (*Sheppard, 2003*; *Van Hooidonk et al., 2016*).

Mayotte (12.1°S, 45.1°E) is the oldest island in the Comoro archipelago, located in the center of the Northern Mozambique Channel. It has somewhat lower coral diversity than surrounding mainland and Madagascar coasts on account of the island area effect (*Obura, 2012*). However, it has among the highest geomorphological diversity of reef habitats in the western Indian Ocean. It is highly eroded with high sedimentation and turbidity in the lagoon (*Thomassin, 2001*). Sea surface temperatures (SSTs) around Mayotte are warm and bimodal (*Ateweberhan & McClanahan, 2010*) determined by the monsoons and interactions of the South Equatorial Current with Madagascar, varying between 25.5 and 29 °C. Coral bleaching on Mayotte has been well documented in 1983 and 1998 (*Faure et al., 1984*; *Quod et al., 2002*), as well as in 2010 (*Eriksson, Wickel & Jamon, 2012*) (*Nicet et al., 2016*, L Bigot & D Obura, 2010, unpublished data).

This study is based on surveys in early June 2010 following the peak months of high sea furface temperatures from February to April (Fig. 1). Bleaching and mortality of corals were observed throughout the island's reefs. This paper focuses on comparisons among coral genera and coral colony size classes in bleaching and mortality patterns, and estimation

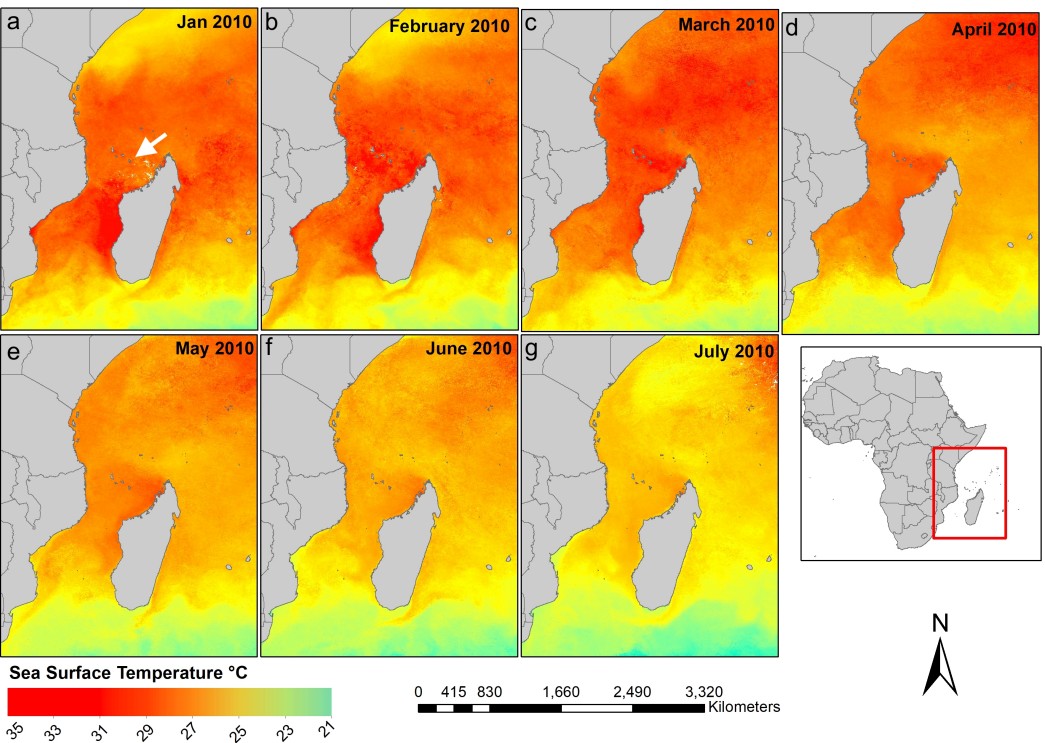

**Figure 1** Sea Surface Temperature (SST) in the Western Indian Ocean, by month, from January to July 2010. Mean monthly level 3 data at 4 km resolution, source: Moderate Resolution Imaging Spectroradiometer (MODIS), https://oceancolor.gsfc.nasa.gov/cgi/l3.

of the full impact of the bleaching event on the corals of Mayotte, in the context of repeat bleaching events.

## METHODS

Coral reef structures on Mayotte Island are diverse, including outer reef banks and slopes, inner slopes of the barrier, a second inner barrier reef at the southwest of the island and fringing reefs around the main island and smaller islets (Fig. 2). The island is densely populated, with high pressure on the lagoon from fishing, and sedimentation from runoff and land-use change (*Thomassin, 2001*; *Bigot et al., 2018*).

Sampling for coral bleaching was conducted on the Tara Oceans Expedition from 30 May–17 June 2010, following peak bleaching months. Twenty seven of the 34 sites (Table 1, Fig. 2) were sampled between a depth of 8 and 12 m, corresponding to the zone of maximum reef development, though on two fringing and one inner barrier sites the reef profile forced sampling to be done at 4–7 m, and on two outer barrier and the two bank reefs, sampling was done at 16–20 m. This paper reports on two survey methods collected as part of a more comprehensive dataset (*Obura & Grimsdith, 2009*). Coral size class structure was sampled with 1 m wide belt transects, including bleaching and mortality observations by colony,

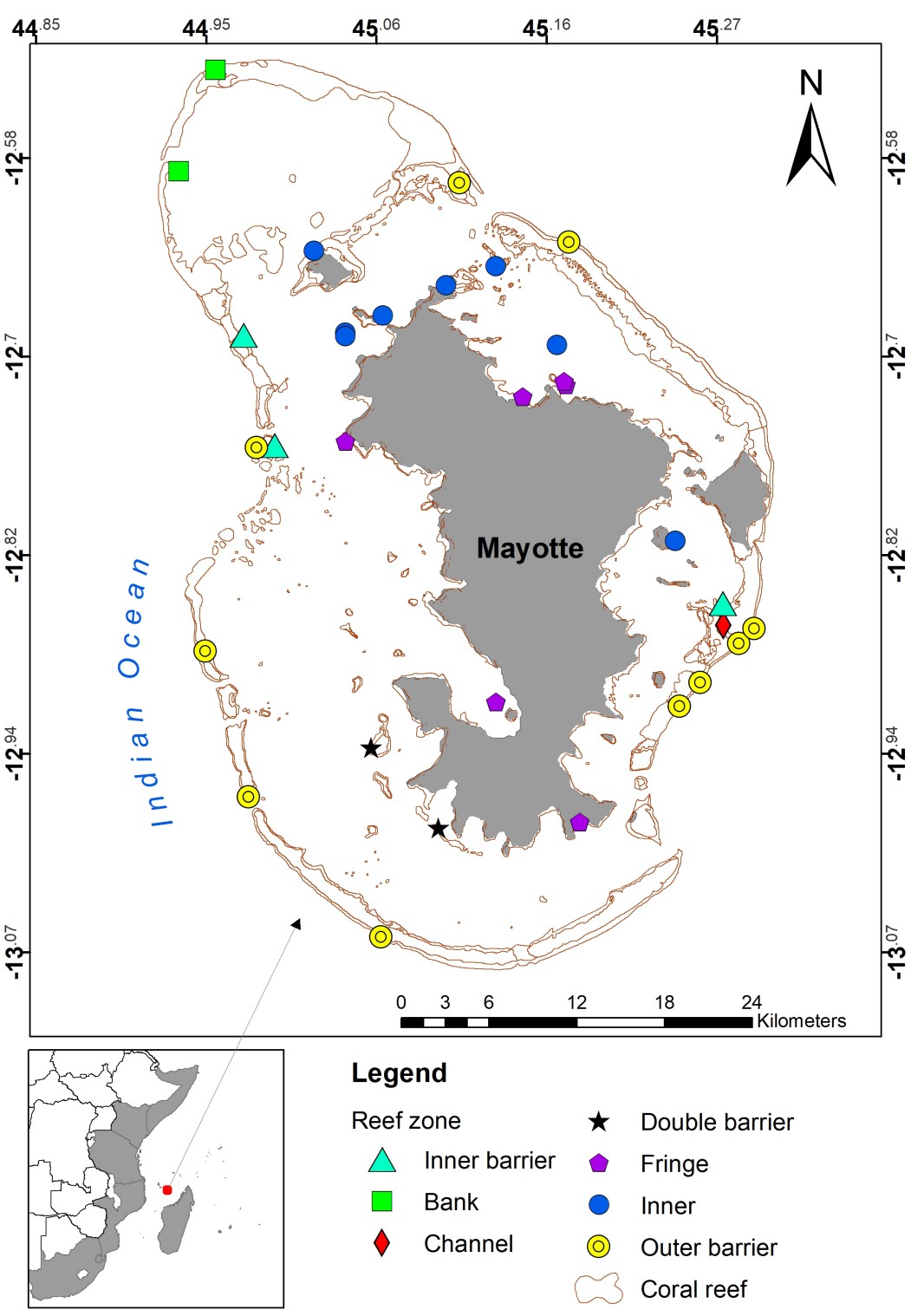

**Figure 2  Map of Mayotte, showing the reef and lagoon structure, and sampling sites coded by reef zone** (*UNEP-WCMC, 2010*).

**Table 1  Sampling details of Tara Oceans Expedition to Mayotte, 2010.** The table shows the depth characteristics of each site, the area of small coral quadrats and large coral transects samples, actual number of coral colonies counted, and standardized number and area of colonies (to 100 m$^2$).

| SITE | Zone | Depth | Area sampled (m$^2$) | | # colonies | Standardized to 100 m$^2$ | |
|------|------|-------|--------|--------|-----------|-----------|-----------|
| | | | <10 cm | >10 cm | counted | # colonies | area (m$^2$) |
| MA05 | bank | 20 | 3 | 12 | 132 | 1,625 | 68.5 |
| MA24 | bank | 20 | 6 | 25 | 97 | 527 | 89.0 |
| MA03 | outer barrier | 10 | 3 | 13 | 299 | 5,691 | 59.8 |
| MA14 | outer barrier | 10 | 3 | 10 | 242 | 3,983 | 48.4 |
| MA15 | outer barrier | 9 | 5 | 20 | 347 | 2,980 | 30.8 |
| MA16 | outer barrier | 10 | 5 | 22 | 303 | 2,366 | 75.2 |
| MA18 | outer barrier | 10 | 3 | 12 | 338 | 5,467 | 70.5 |
| MA22 | outer barrier | 11 | 6 | 25 | 208 | 1,529 | 42.8 |
| MA27 | outer barrier | 16 | 6 | 25 | 217 | 1,349 | 30.7 |
| MA28 | outer barrier | 16 | 6 | 25 | 328 | 2,832 | 19.6 |
| MA29 | outer barrier | 12 | 6 | 25 | 315 | 2,666 | 20.2 |
| MA30 | outer barrier | 12 | 6 | 25 | 357 | 2,644 | 24.4 |
| MA19 | channel | 10 | 5 | 20 | 256 | 2,165 | 27.0 |
| MA08 | inner barrier | 10 | 4 | 20 | 341 | 3,045 | 40.2 |
| MA10 | inner barrier | 10 | 3 | 11 | 248 | 4,824 | 36.7 |
| MA32 | inner barrier | 6 | 6 | 25 | 186 | 1,175 | 90.3 |
| MA12 | double barrier | 8 | 4 | 15 | 173 | 1,923 | 27.6 |
| MA13 | double barrier | 10 | 3 | 10 | 190 | 2,880 | 47.6 |
| MA04 | inner | 10 | 4 | 20 | 246 | 2,430 | 43.8 |
| MA06 | inner | 10 | 5 | 20 | 305 | 2,665 | 72.6 |
| MA07 | inner | 10 | 5 | 20 | 361 | 2,945 | 41.0 |
| MA17 | inner | 9 | 6 | 25 | 208 | 1,351 | 17.9 |
| MA20 | inner | 8 | 6 | 25 | 218 | 1,404 | 48.2 |
| MA23 | inner | 11 | 6 | 25 | 170 | 1,643 | 22.4 |
| MA25 | inner | 10 | 6 | 25 | 513 | 3,673 | 36.4 |
| MA26 | inner | 11 | 6 | 25 | 236 | 1,932 | 10.5 |
| MA34 | inner | 9 | 12 | 50 | 308 | 1,028 | 31.3 |
| MA01 | fringe | 10 | 2 | 7 | 112 | 2,243 | 41.5 |
| MA02 | fringe | 10 | 6 | 25 | 144 | 1,083 | 14.2 |
| MA09 | fringe | 10 | 6 | 25 | 209 | 1,710 | 13.0 |
| MA11 | fringe | 8 | 5 | 23 | 144 | 1,049 | 32.2 |
| MA21 | fringe | 8 | 6 | 25 | 231 | 1,076 | 25.8 |
| MA31 | fringe | 7 | 6 | 25 | 173 | 1,135 | 28.8 |
| MA33 | fringe | 4 | 6 | 25 | 284 | 2,073 | 74.3 |
| | *Totals* | | | | | *Averages:* | |
| | 176 | | 730 | | 8,439 | 2,327 | 41.3 |

and visual estimation of abundance of coral genera on a 1–5 scale, and of percent coral cover.

Belt transects 1 m wide were used for sampling coral colony sizes. Coral colonies whose center fell within the belt and quadrats) were counted. The largest colony diameter was recorded in the size class bins: 11–20, 21–40, 41–80, 81–160, 161–320 and >320 cm. For corals smaller than 10 cm, subsampling was done using 1 m$^2$ quadrats at the 0, 5, 10, 15, 20, and 25 m transect marks, in size class bins 0–2.5, 3–5 and 6–10 cm. Colony condition was recorded as unaffected (no visible effect of thermal stress), pale, bleached, partially dead or fully dead, as exclusive categories by colony. Colonies were assigned to the most severe condition observed if that condition occupied more than 1/5 of the colony surface; i.e., a colony with bleaching and partial mortality was classed as partially dead if more than about 1/5 was dead. A 1 m stick was used to help guide estimation of transect width and mark the 1 m$^2$ quadrats, and the stick was marked at 10, 20, 40 and 80 cm to guide size estimation of coral colonies. A standard transect length of 25 m was targeted, but length was often limited by a high density of corals and time available per dive. Seventeen of the 34 sites were sampled with 25-m transects (Table 1). The smallest sampling was of two 1 m$^2$ quadrats and a 7-m belt transect, recording 18 small and 94 large corals, respectively. The minimum counts of 11 small and 86 large corals were recorded within a 25-m transect.

Sampling focused on coral genera that were already known to cover a range of bleaching susceptibility from high to low and that are generally common on East African reefs (*Obura & Grimsdith, 2009*): 1—low resistance to bleaching: *Acropora* (including *Isopora), Montipora, Pocillopora, Seriatopora, Stylophora*; 2—intermediate resistance to bleaching: *Echinopora, Dipsastraea, Favites, Goniastrea, Leptastrea, Platygyra, Acanthastrea, Coscinaraea, Fungia, Galaxea, Hydnophora, Lobophyllia, Oxypora, Pavona, Plerogyra* ; 3—high resistance to bleaching: *Porites* (massive and branching morphologies recorded separately) and *Turbinaria*.

The relative abundance of all coral genera at a site was recorded by visual estimate, on a five–point scale (rare, uncommon, common, abundant, dominant) following *Devantier & Turak (2017)*. An index of relative abundance for each genus was calculated from these estimates as the average of: the proportion of sites at which a genus was present, its average abundance across all sites, and maximum abundance at any site, all converted to 0–5 scale (*Obura & Grimsdith, 2009*).

For analysis of the coral size class data, all densities per genus were transformed to a standard area of 100 m$^2$. The number and area of colonies per 100 m$^2$ was used. Colony area was calculated for each size class using its median diameter and assuming the area of a coral colony is approximated by an ellipse with the second diameter half of the maximum (area = 1/2*pi*r$^2$). Proportions of each of the colony condition classes were calculated based on abundance and area of each in each size class. For some analyses the number of classes was reduced to three, by aggregating pale and bleached (termed bleached) and partially and full dead (termed dead).

To investigate the impact of different approaches to monitoring and reporting bleaching, two variations of the bleaching and mortality data were analyzed. First, we tested the effect of excluding the proportion of normal colonies, because many programmes count or report

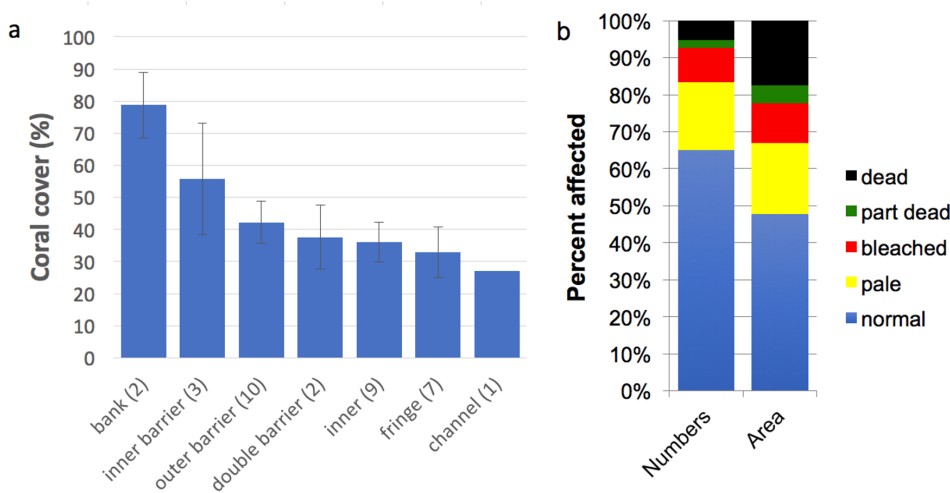

**Figure 3** **Coral cover and condition in Mayotte, June 2010.** (A) Coral cover by reef zone (mean ± se). The number of sites sampled in each zone is shown in parentheses in the *x* axis labels. (B) Proportional bleaching and mortality of corals in Mayotte, in June 2010, by number and area of colonies sampled.

only colonies showing some level of bleaching and/or mortality, and not count the number of normal corals. Second, we tested the effect of weighting conditions of increasing severity, i.e., from pale, to bleached, to partially dead to dead (see *McClanahan et al., 2007b*). We applied the following weights: 1*pale, 2*bleaching, 3*partial mortality and 4*mortality. Cluster analysis using the Bray-Curtis similarity index was found to give results easy to interpret for the pros and cons of the different approaches. Analysis was conducted in PRIMER v6.0 with significant groupings identified using SIMPROF (*Clarke & Warwick, 2001*).

Estimates of the potential final mortality of corals were obtained from the size class data based on the assumption that minimum mortality from the event would be equivalent to current levels of partial and full mortality. This was obtained by subtracting the sum of partial and full mortality from the total counts for all size classes for each genus. By contrast, the assumption that all currently bleached corals would die gives an estimate of maximum mortality from the event, thus subtracting the counts of bleached corals plus partial and fully dead corals. We assumed that pale corals survived.

## RESULTS

Thirty-four sites were surveyed overall, in seven reef zones (Table 1, Fig. 2). In total, 8,439 colonies were sampled. Coral cover varied from a maximum of almost 80% on the offshore Banc d'Iris, between 35–45% for outer barrier, inner barrier and small-island reefs within the lagoon, slightly over 30% for fringing reefs on the main island, and <30% in the Passe en 'S', the only channel site surveyed (Fig. 3A). Nevertheless, coral cover was not significantly different by reef zone (One Way ANOVA, ($F = 1.338$, $P = 0.275$). No patterns were observed, nor have been reported in the literature, of differential coral genus distributions around the island unrelated to reef zone, so all sites were lumped together in

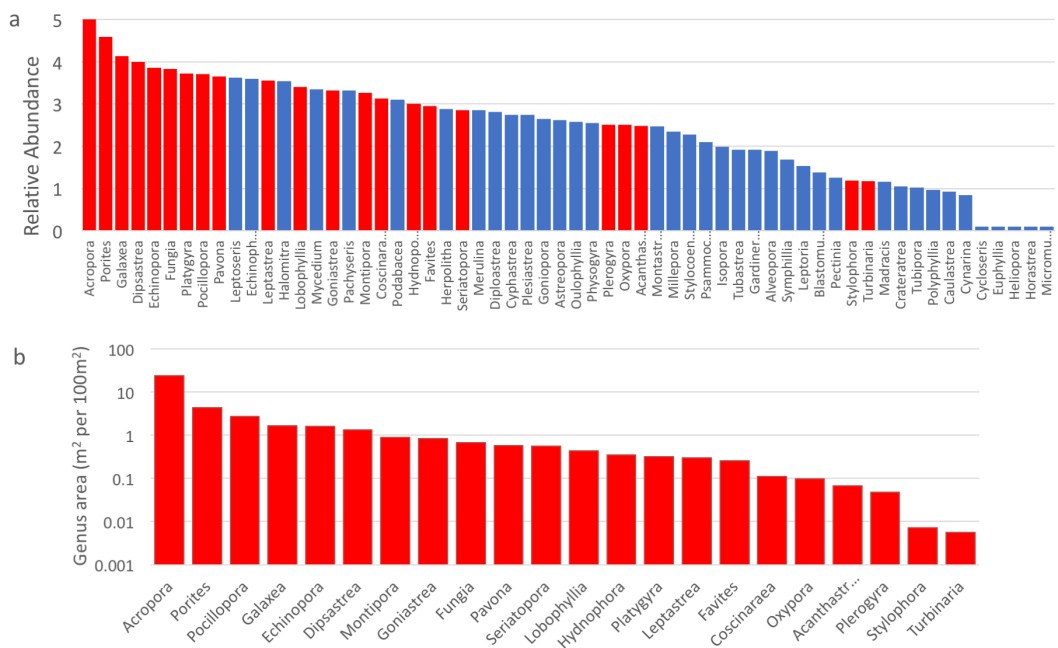

**Figure 4 Coral genera abundance, Mayotte, 2010.** (A) Relative abundance of all coral genera identified, aggregated across sites. The index of relative abundance ranges from a maximum of 5, with zero representing absence. Site level scores were: rare (1), uncommon (2), common (3), abundant (4) and dominant (5). (B) Area (biomass) of each genus measured in size class transects (in $m^2$ per 100 $m^2$ of reef area), ranked from highest to lowest. Genera sampled in both methods (A and B) shaded red, genera sampled only in visual estimates (in A) shaded blue.

subsequent analyses, to focus on patterns in bleaching by coral traits (taxonomy, colony size). Across all sites, average coral colony abundance was 2327 ($\pm$1,253) and area was 41 $m^2$ ($\pm$21.9) per 100 $m^2$ of reef. By abundance, 35% of coral colonies were affected by the bleaching event, with 18.4% being pale, 9.3% bleached, 2% partially dead and 5.3% dead (Fig. 3B). By area, 52% of coral area or biomass was affected by bleaching, with 19.3% being pale, 10.7% bleached, 4.8% partially dead and 17.5% dead.

A total of 60 coral genera were recorded (Fig. 4A), with clear dominance by *Acropora*, followed by *Porites*. All of the most abundant nine genera, and 17 of the top 25, were among those targetted for size class sampling (Fig. 4B). Genera omitted from size class sampling, but that were present at moderate levels of abundance included *Leptoseris* (rank = 10), *Echinophyllia* (11), *Halomitra* (15), *Mycedium* (17), *Pachyseris* (18) and *Podabacia* (20). *Acropora* contributed 58% of the total area of sampled corals (Fig. 4B), about five times greater than *Porites*, the second genus. Sixteen genera were recorded at abundances between 1–10 $m^2$ per 100 $m^2$ of reef, with four at low abundance between 0.1–1 $m^2$, and two at very low abundance (*Stylophora* and *Turbinaria*). For the genera sampled using both methods, the logarithm of the area sampled and their visually assessed relative abundance index were strongly correlated ($r^2 = 0.894$), though some genera switched ranks between the two methods (Fig. 4).

Bleaching and mortality varied widely across genera, from 100% bleached in *Turbinaria* to 6% pale in *Acanthastrea*. Genera with fewer than 20 colonies sampled (i.e., *Turbinaria*, *Oxypora*, and *Stylophora*) are excluded from further analysis, so the following results are for the 20 remaining genera. The most susceptible genera to combined bleaching and mortality (Fig. 5A) were *Pocillopora* and *Montipora* (>70% total impact) followed by *Lobophyllia*, *Porites* (massive species), *Dipsastrea*, *Goniastrea*, *Acropora* and a range of others (40–65%). *Leptastrea* and *Acanthastrea* showed lowest levels of combined pale and bleached colonies (≈6%), with no mortality. *Pocillopora* and *Acropora* were the only genera with high levels of mortality (47% and 32%, respectively, combined partially dead and dead), though mortality was also observed in *Porites* (massive and branching species), *Echinopora*, *Favites*, *Seriatopora*, *Platygyra*, and *Hydnophora*. *Pocillopora* showed the most extensive and complex response (18% pale, 14% bleached, 21% partially dead and 26% dead), while *Acanthastrea* was the least impacted (6% pale). Most genera showed a higher degree of paling (up to 50% in *Dipsastraea*) than full bleaching (up to 31% in *Montipora*). Simplified into three categories (unaffected, bleached and dead) and illustrated in a ternary plot, *Acropora* and *Pocillopora* showed intermediate levels of all three (Fig. 5B). Other genera occupy a zone near the axis marking low mortality, at varied levels of unaffected and bleached.

Cluster analyses showed the impact of excluding unaffected colonies, and the use of weighting coefficients (Fig. 5C). Without weights, including 'unaffected' as a category distinguished three significant clusters (case 1): (a) *Acropora* and *Pocillopora* due to their high combined bleaching and mortality, (b) a large group of intermediate genera showing some bleaching and limited mortality, and (c) a smaller group of five genera mostly affected by bleaching, but at lower abundance. Excluding 'unaffected' (case 2) shifted the boundary between (b) and (c) such that (c) was reduced to two genera (*Leptastrea* and *Acanthastrea*), the other three genera shifting into group (b). Using weighting coefficients and including 'unaffected' (case 3) altered case 1 by splitting the five low-response genera into two groups—the three that were pushed into group (b) in one group, and *Acanthastrea* and *Leptastrea* in a separate group. Finally, using weighting coefficients but excluding 'unaffected' (case 4) put all of the genera except *Acanthastrea* and *Leptastrea* into a single group, and these two in their own small group.

Two conclusions can be drawn from the above. First, that including normal or 'unaffected' colonies in field counts and analysis is important, as cases 2 and 4 both showed less discrimination of bleaching responses (less discrimination of the first order spatial patterns in Fig. 5B) than cases 1 and 3, respectively. Second, that including weighting coefficients alters the results, but varying with the set of variables included. Comparing cases 2 and 4, adding weighting coefficients worsened the result, removing the distinction between *Acropora* and *Pocillopora* (bleaching with mortality) from the main group of genera (paling and bleaching, no mortality). Comparing cases 1 and 3, adding weighting coefficients added resolution in the low-response low-abundance group, distinguishing *Leptastrea* and *Acanthastrea* (with only minor paling and bleaching, <6%, Fig. 5A) from *Porites* (branching), *Favites* and *Seriatopora* (from 20–33% impacted, and varied amounts of paling, bleaching and mortality). While the latter result adds value to interpretation,

**a**

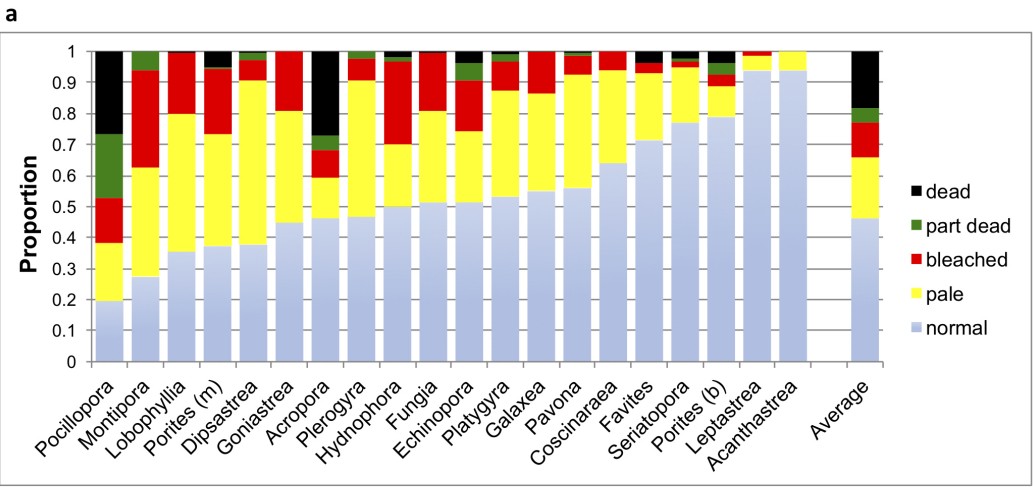

**b**

**c**

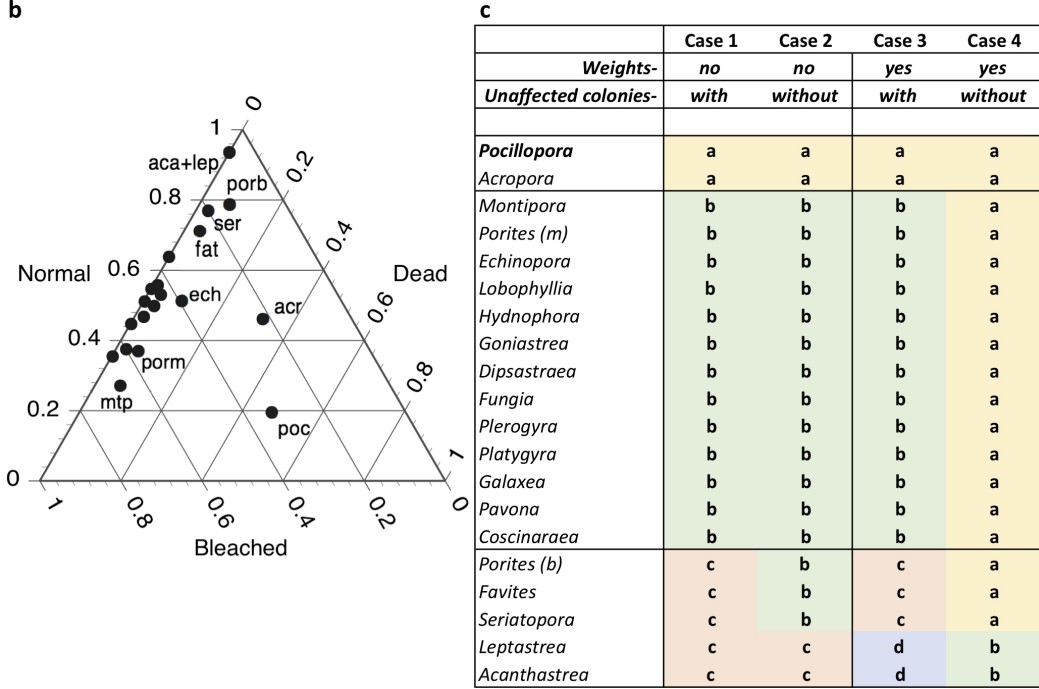

| | Case 1 | Case 2 | Case 3 | Case 4 |
|---|---|---|---|---|
| *Weights-* | *no* | *no* | *yes* | *yes* |
| *Unaffected colonies-* | *with* | *without* | *with* | *without* |
| | | | | |
| *Pocillopora* | a | a | a | a |
| *Acropora* | a | a | a | a |
| *Montipora* | b | b | b | a |
| *Porites (m)* | b | b | b | a |
| *Echinopora* | b | b | b | a |
| *Lobophyllia* | b | b | b | a |
| *Hydnophora* | b | b | b | a |
| *Goniastrea* | b | b | b | a |
| *Dipsastraea* | b | b | b | a |
| *Fungia* | b | b | b | a |
| *Plerogyra* | b | b | b | a |
| *Platygyra* | b | b | b | a |
| *Galaxea* | b | b | b | a |
| *Pavona* | b | b | b | a |
| *Coscinaraea* | b | b | b | a |
| *Porites (b)* | c | b | c | a |
| *Favites* | c | b | c | a |
| *Seriatopora* | c | b | c | a |
| *Leptastrea* | c | c | d | b |
| *Acanthastrea* | c | c | d | b |

**Figure 5** **Bleaching and mortality of coral genera.** (A) Proportion of bleaching and mortality by genus, excluding the two least abundant genera. (B) Ternary plot of unaffected, bleached (pale plus bleached) and dead (partial plus full mortality) for genera sampled in the study. Key genera are indicated using three-letter codes that correspond to the first three letters of genus names in (A) and (C), except for 'fat' (*Favites*), 'mtp' (*Montipora*), 'porb' (branching *Porites*) and 'porm' (massive *Porites*). (C) Cluster analysis results with SIMPROF test to show significant clusters of genera at $p = 5\%$ level for four cases: unweighted proportions of pale, bleached, partial and full mortality analyzed with (case 1) and without (case 2) the proportion of unaffected colonies; and with weights applied to bleached (x2), partial mortality (x3) and full mortality (x4) (see methods), also with (case 3) and without (case 4) the proportion of unaffected colonies. Letters show significant groups within each test.

weighting should be used with caution and tested on each dataset to which it is applied, to fully understand how it influences the results.

By area, the coral community was dominated by mature colonies in the 81–160 cm size class (Fig. 6A) followed by younger colonies from 11–80 cm and then the largest colonies above 1.6 m. Numerically, 6–10 and 11–20 cm corals were most abundant. Bleaching and mortality varied by colony size (Fig. 6B). Both bleaching and mortality showed a strong linear increase with coral colony size, up to 1.6 m. Of the smallest colonies only 17% were affected by bleaching, with 3% suffering mortality. These proportions increased progressively to the most impacted size class, 81–160 cm, for which 67% of colonies were affected with 35% mortality. The sample size for corals >160 cm was low, with 20 and 1 in the 160–320 and >320 cm size classes, respectively, compared to from 131 to 4257 colonies in the smaller size classes. For corals <10 cm, the ratio of mortality to bleaching was <0.2, which increased to 0.3–0.5 for intermediate colonies from 10–80 cm. For larger corals, 81–160 and 161–320 cm, mortality exceeded bleaching, with ratios of 1.6 and 1.1 respectively.

# DISCUSSION

Bleaching and mortality averaged 30 and 22%, respectively, across all sites (Fig. 3B) though with high levels of variability from near zero at sites in the south to maximum mortality levels at sites in the north and east of Mayotte. These results agree largely with *Eriksson, Wickel & Jamon (2012)* who found 10% bleached and 40% dead corals on the more highly impacted northern and eastern reefs during May 2010. Percent coral cover varied over a wide range across reef zones in Mayotte (Fig. 3A) though without statistical significance. Accordingly, we aggregate the coral community of Mayotte to analyze variance in bleaching response among genera.

## Do genus and size affect coral bleaching?

*Pocillopora* and *Acropora* were the only coral genera to show significant mortality as a result of the 2010 event, setting them apart from the others as the most susceptible genera to bleaching, in accordance with findings throughout the Indo-Pacific (*Marshall & Baird, 2000*; *Obura, 2001*; *Loya et al., 2001*; *McClanahan et al., 2004*; *Van Woesik et al., 2012*). Cluster analysis of the bleaching responses identified two additional groups of species—a large group with intermediate responses characterized by variable but low mortality and variable levels of pale and bleached colonies (group 'b', case 1, Fig. 5C), and a smaller group with low levels of bleaching and variable but low mortality (group 'c'). The middle group included a wide range of genera including *Porites* (massive species), various merulinids, agariciids, siderastreids, and fungiids. The last group included *Porites* (branching species), *Favites*, *Seriatopora*, *Acanthastrea*, and *Leptastrea*. These groups are in broad agreement with previous reports from East Africa (e.g., massive *Porites* is usually ranked among the least susceptible to bleaching and mortality, branching *Porites* among the most susceptible— *Obura, 2001*; *McClanahan, 2004*). We have not analyzed the symbiont characteristics of genera sampled here, which has significant effects on bleaching susceptibility and mortality, particularly when different host-symbiont combinations may be possible (e.g., see

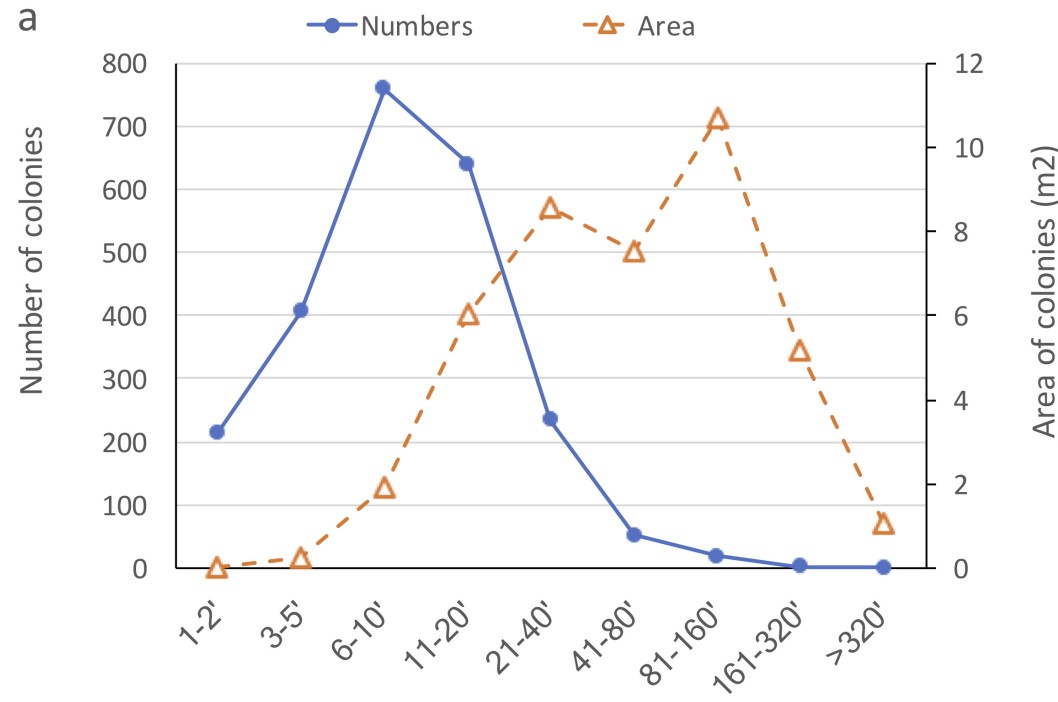

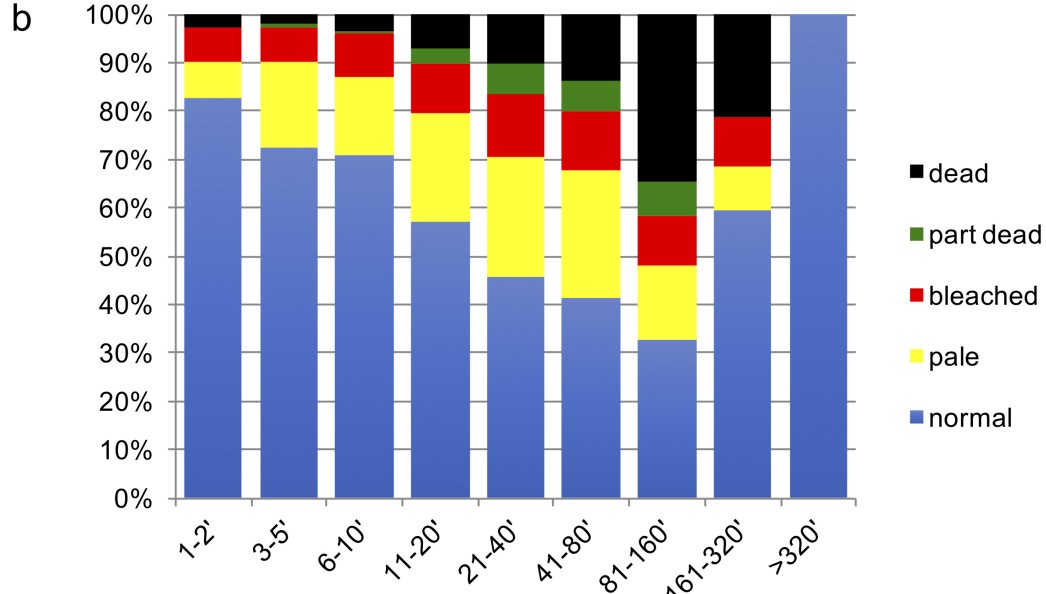

**Figure 6 Bleaching and mortality of corals, by colony size.** (A) Size class distributions of corals from all sites sampled, by number of colonies (left axis, closed circles) and area (in m$^2$) of colonies (right axis, open triangles) per 100 m$^2$ of reef area. (B) Overall bleaching and mortality proportions by colony area in each size class . Data is combined across all coral genera sampled in Mayotte, in June 2010.

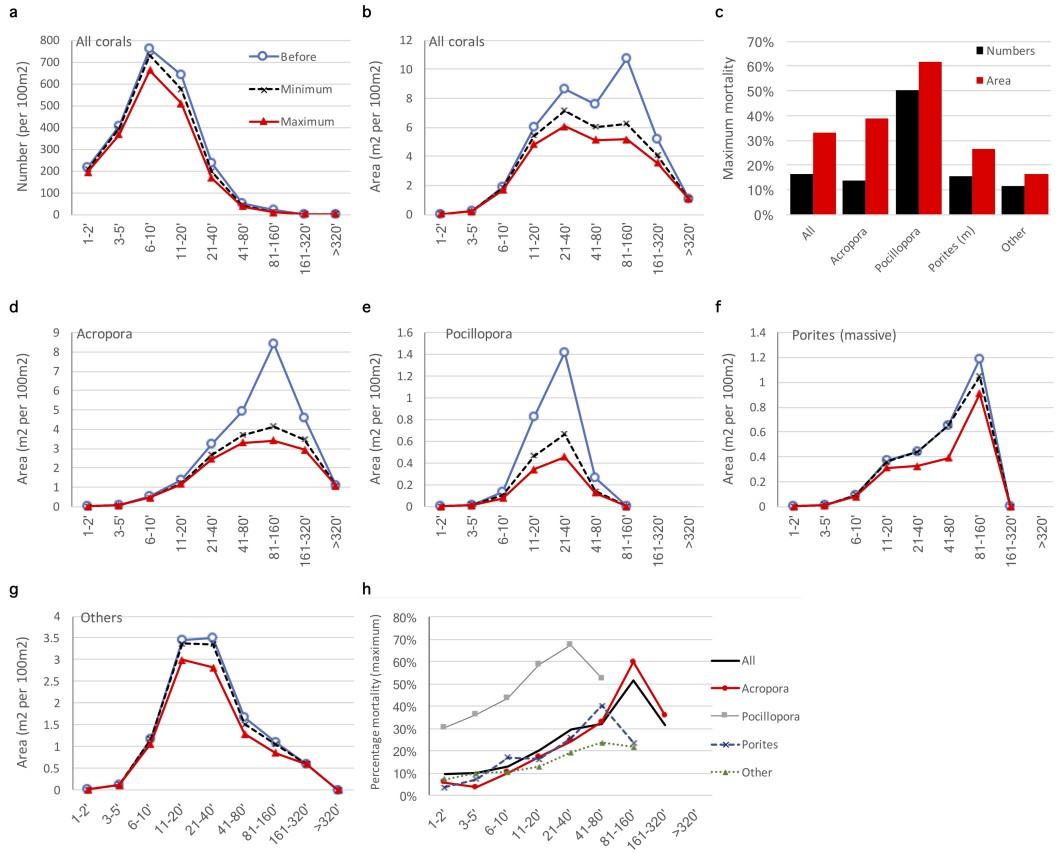

**Figure 7** **Estimates of the minimum and maximum impact of the 2010 coral bleaching event on the coral community of Mayotte, 2010.** Size class distribution of all corals by (A) abundance and (B) area in each size class. The following results are presented for the key taxa *Acropora*, *Pocillopora*, *Porites* (massive species) and other genera combined: (C) estimated maximum percentage loss of corals, by number and area of colonies; (D–G) estimated minimum and maximum percentage loss by area in each size class; (H) estimated maximum percentage loss of corals by size class. Legend in plot A applies to plots B and D–G.

*Stat et al., 2009*; *Baker & Romanski, 2007*; *Oliver & Palumbi, 2009*). Overall, though this appeared to be a severe bleaching event for reefs in Mayotte due to the dominance of *Acropora* and consequent loss of total coral cover, and visual dominance of bleached and dead *Acropora* and *Pocillopora*, the thermal stress to other genera was not sufficient to result in significant mortality.

A clear size differential in bleaching and mortality susceptibility was found, with susceptibility being greatest in large corals. For corals in the three smallest size classes (Fig. 6B) the proportion of bleached colonies increased (15 to 25%) with size, with little increase in the proportion of mortality (<5%). For adult corals from 10 cm and above, the proportion of bleached corals remained relatively stable (25%), but the proportion of mortality increased (10 to 40%), and exceeded the proportion bleached. This response was strongly determined by the size-dependent response of *Acropora*, but it did occur in other genera (Fig. 7H). Lower bleaching levels in small colonies, and particularly recruits,

has been noted in other locations and coral taxa, particularly *Oculina patagonica* in the Mediterranean (*Shenkar, Fine & Loya, 2005*), and three dominant species in the Florida Keys (*Colpophyllia natans, Montastrea faveolata*, and *Siderastrea siderea,* (*Brandt, 2009*). However, in a mild bleaching event *Ortiz, Del C Gomez-Cabrera & Hoegh-Guldberg (2009)* found no relation of size to bleaching extent, and *Bak & Meesters (1999)* predicted that bleaching impacts would be selectively higher on small rather than large corals.

From a methodological point of view, these results suggest several important considerations for measuring the impact of bleaching events at colony level, and extrapolating to community level indices. First, colony size is important in the bleaching susceptibility of corals for two reasons—smaller corals are less susceptible to bleaching, and larger corals have an exponentially greater contribution to biomass and area. Second, it is important to include unaffected colonies in the counts, as this improves identification of differential bleaching responses (Fig. 5C), and without total numbers of colonies the prevalence of bleaching and mortality in relation to the total population cannot be determined. Third, a corollary of the first two, is the importance of an unbiased sample of all corals, unaffected and bleached, and across all size classes. Without a strict fixed-area sampling method, such as with a physical transect or quadrat as a guide, unconstrained or haphazard counts are likely to both (a) oversample colonies that show a response over unaffected ones, as observers will tend to count what they are looking for (bleached and dead corals) and many normal colonies are brown and inconspicuous, and (b) small colonies will always be under-sampled compared to larger colonies, particularly when including sizes under 10 cm. Fourth, differential weighting of bleaching and mortality categories can have a strong influence on results, so should be done with caution in each case. In our results it improved discrimination of low-response groups when all colonies (impacted and unaffected) were included, but when unaffected corals were excluded it worsened the result. Thus where sampling is not restricted by physical quadrats or transects, and the size of corals is not measured (one and three above), the impact of weighting on results may not be possible to ascertain.

## Estimating the impact of a bleaching event

The surveys took place in early June, over 1 month after the end of peak temperatures at the end of April 2010 (Fig. 1), so this dataset likely presents peak levels of combined bleaching and mortality. This is a common challenge in interpreting the eventual impact of a bleaching event, as surveys are often targeted for peak bleaching conditions, but unless follow up surveys are done, it cannot be known if bleached corals recovered or died. However, the envelope of possible outcomes of a bleaching event can be estimated from peak bleaching levels assuming further mortality from bleaching is either zero or maximal. That is, currently bleached corals all recover, or die, respectively.

The present dataset allows this to be done across coral size classes and genera, to estimate minimum and maximum potential impact of the bleaching event (Fig. 7). By number of colonies the minimum and maximum loss of corals appears minor (Fig. 7A) but is significant by area (Fig. 7B), particularly for 81–160 cm corals. Maximum mortality by colony abundance is estimated at 10–15% for *Acropora*, *Porites* (massive) and other

genera (Fig. 7C), but at 50% for *Pocillopora*. By contrast, maximum mortality by colony area is estimated at 40% for *Acropora*, 25% for *Porites* (massive), <20% for other genera (Fig. 7C), but at >60% for *Pocillopora*. By colony abundance, maximum mortality overall is estimated at 16%, but by colony area, 32%, a major difference in result affected by consideration of colony area. This is because the bleaching event preferentially impacted larger colonies, paticularly *Acropora*, eliminating the dominance of 81–160 m colonies (Fig. 7D). Reefs in Mayotte are strongly dominated by staghorn and tabular *Acropora* species, and dominance of the 81–160 cm size class in April suggests the community was approaching maturity where the community would be dominated by even larger stands of tabular and staghorn *Acropora*. The bleaching event strongly flattened the size class distribution with similar area in 41–320 cm classes after the event.

*Pocillopora* populations were strongly dominated by 21–40 cm and 11–20 cm colonies pre-bleaching (Fig. 7E), providing the corresponding peak in the overall population curve (Figs. 5A, 6B); their area was reduced by about 60% by the bleaching event (Fig. 7C), though they still remained dominant over other size classes of *Pocillopora*. The population of massive *Porites* was strongly dominated by 81–160 cm corals, which remained dominant after bleaching. The remaining genera were most strongly represented by 11–40 cm colonies, and mortality was relatively minor and evenly spread across them, maintaining the same size class distribution following bleaching. Finally, estimated maximum mortality increased with size for all corals and for the three key genera *Acropora*, *Pocillopora*, and *Porites* (Fig. 7H). Interestingly, these three genera all show a decline in mortality for their largest size class.

## Recurring bleaching in Mayotte

The coral reefs of Mayotte have been impacted by multiple significant bleaching events. In May–June 1983 bleaching was documented at 0–18% for fringing reefs, 30–45% for lagoon reefs and 30–75% for outer barrier reefs, though an indication of the final mortality was not noted (*Faure et al., 1984*). In 1998, 36% of the reefs had not recovered from the 1983 bleaching event, and mortality of >80% of *Acropora* tables on outer reef slopes was reported from April to August (*Quod et al., 2002*), which judging by patterns in this study may have reflected approximately 50% mortality including inner reefs. The 2010 event recorded here resulted in 32% mortality of all corals, and just under 40 % for *Acropora* alone (Fig. 7C). Observations in 2016 suggest that mortality of corals was between 25 and 50% using the same methods used here (by area, D Obura, pers. obs., 2016), and was recorded at 10–30% by colony number using other methods (*Nicet et al., 2016*), so roughly comparable to the 2010 event.

In all four events, higher impact occurred on outer barrier reefs and lower levels in the lagoon and on fringing reefs, and the most dramatic bleaching and mortality was of tabular and staghorn *Acropora* colonies on outer reefs. Because of its dominance of the coral community, *Acropora*'s response to thermal stress dominated the overall community response. The intervals between these major bleaching events has progressively declined—from 16 years to ten and six years. Across all of these events the reefs have apparently achieved considerable recovery and maintained the same dominance by *Acropora*. This

suggests a high degree of community resilience, likely partly a result of high levels of connectivity (*Crochelet et al., 2016*) due to the complex eddies that maintain high self-seeding of reefs within the northern Mozambique channel (*Obura et al., 2018*; *Bigot et al., 2018*).

However, anthropogenic stresses in Mayotte from both fisheries and water quality degradation (*Wickel & Thomassin, 2005*) are increasingly evident in the greater prevalence of coral disease and chronic mortality from unknown sources (D Obura, pers. obs., 2016). This will likely undermine the natural resilience of the reefs (*Obura, 2005*; *Hughes et al., 2010*) and may reduce their ability to recover from the 2016 bleaching event.

Between latitude S 12 −13.5 and longitudes E 44.5−45.5 (a box around Mayotte) SST has warmed by 0.096 °C per decade for the thirty-year period from 1981–2010 (*Reynolds et al., 2002*), slightly less than the global level of 0.147 °C per decade (*Rayner et al., 2003*). Mayotte is in the region with the lowest SST rise in Eastern Africa (*McClanahan et al., 2007a*). *Donner (2009)* estimated a rate of SST rise to which corals must adapt to avoid catastrophic coral decline, of 1.5 °C in 50–80 years. This is 0.2–0.3 °C per decade, some two to three times higher than the rise in temperatures that coral reefs in Mayotte have experienced over the course of four bleaching events. While Mayotte's reefs have shown remarkable resilience so far, it is not clear that they are acclimating or adapting sufficiently to the rise experienced of 0.1 °C per decade. Further, the shorter intervals for recovery between events, matching the global pattern now at sub-decadal levels (*Hughes et al., 2018*), is approaching limits for recovery consistently used in framing the onset of 'catastrophic' bleaching (*Sheppard, 2003*; *Van Hooidonk et al., 2016*).

## CONCLUSION

The most recent analysis identifies 2030 as approximately the year in which Mayotte will experience Annual Severe Bleaching under RCP 8.5 (business as usual scenario, equivalent to today's CO2 emission rates; *Van Hooidonk et al., 2016*). That the reefs are already experiencing decadal severe bleaching only 15–20 years earlier is strong indication that the trajectory for coral reefs in the region towards decline may be inexorable on the time scales at hand, and that the reefs cannot withstand annual occurrence of the scale of bleaching documented in 1983, 1998 and 2010, and repeated in 2016 (*Nicet et al., 2016*). Although the Northern Mozambique Channel may be a center of diversity and of key significance to the Western Indian Ocean at large (*Obura, 2012*; *Obura et al., 2018*), it may not be a refuge from warming for coral reefs (*McClanahan et al., 2014*) and urgent and emergency planning is needed to identify what can be done to secure the best possible future not just for the reefs of Mayotte or the WIO, but also more broadly on a global scale (*Beyer et al., 2018*).

## ACKNOWLEDGEMENTS

We are grateful to Etienne Bourgois, the Tara schooner and its captain Hervé Bourmand, Mathieu Oriot (Diving Officer), the Tara crew, the Oceans Consortium, and James Mbugua. Tara Oceans would not exist without continuous support from 23 institutes

(http://oceans.taraexpeditions.org). This article is contribution number 77 of the Tara Oceans Expedition 2009–2012.

### Funding

Funding for David Obura was provided by the Western Indian Ocean Marine Science Association, Grant No: MASMA/OR/2008/05. The funders had no role in study design, data collection and analysis, decision to publish, or preparation of the manuscript.

### Grant Disclosures

The following grant information was disclosed by the authors:
Western Indian Ocean Marine Science Association: MASMA/OR/2008/05.

### Competing Interests

The authors declare there are no competing interests.

### Author Contributions

- David O. Obura conceived and designed the experiments, performed the experiments, analyzed the data, contributed reagents/materials/analysis tools, prepared figures and/or tables, authored or reviewed drafts of the paper, approved the final draft.
- Lionel Bigot performed the experiments, contributed reagents/materials/analysis tools, authored or reviewed drafts of the paper, approved the final draft.
- Francesca Benzoni conceived and designed the experiments, approved the final draft, expedition leader and funding.

### Data Availability

    The raw data are provided in a Supplemental File.

### Supplemental Information

Supplemental information for this article can be found online at http://dx.doi.org/10.7717/peerj.5305#supplemental-information.

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
