# Peer review of "Coral responses to a repeat bleaching event in Mayotte in 2010"

_PeerJ, doi:10.7717/peerj.5305_

## Round 0.1 · original submission · Minor Revisions

Three expert reviewers have evaluated your manuscript and their comments and an annotated PDF have been made available. I am aware that there is a potential conflict of interest with one of the reviewers and I have taken this into account in making my decision. All 3 evaluations are consistent and the reviewers have supplied thoughtful and constructive reviews. All suggest minor revisions. I agree with this assessment and encourage you to take into account their comments and suggestions in preparing a new version of the manuscript.

·

Basic reporting

Minor editing and modification of figures needed as detailed in annotated pdf of manuscript. Paper largely clear and well-written. Figures are relevant and well designed but in need of a few edits.

Experimental design

The design of the data collection and analysis meet standards of this research community. Analyses and conclusions drawn from them are mostly appropriate, although one analysis (with and without unaffected) should be removed as no justification is given for why unaffected values should ever be excluded from the analyses.

Validity of the findings

Data, analyses, and findings are robust and sound except as mentioned above.

·

Basic reporting

This is a well-written and analyzed assessment of a 2010 bleaching event in Mayotte. The writing is clear and there is sufficient reference to supporting and relevant literature in the field that places this study in the context of others. Similarly, the authors have indicated the contribution that their findings making to the gaps in the literature.

The authors report on the statistics of the bleaching event from a wide array of transect and quadrat data and report their findings in terms of both genera and area of the reef.

I think the most significant impact of this study is their use of cluster analysis (fig 7c) to assess the changing outcomes when unaffected colonies are/are not included and when weighted components are included. This was a really interesting and useful outcome of the data presented, particularly in terms of application to other reef locations affected by bleaching around the world. I recommend that these findings be incorporated into the abstract in a more definitive way. This part of the study seems to be "undersold" to me and could be incorported more thoroughly.

Experimental design

The overall design and explanation of the study is complete and easy to follow and understand. The statistics used are appropriate.

The study fulfills a gap in knowledge related to reporting and assessment of bleaching events and the susceptibility of genera in the event. An assessment of a past bleaching event (2010) is especially timely as we (quickly) approach greater frequency of these events.

Validity of the findings

The authors follow the standards of Peer J in terms of a robust data set that has been statistically analyzed and assessed. The authors do not indicate the impact or novelty of their results, nor do they overly speculate on their findings, however, their conclusions are well stated and associated with their overall research question.

My only suggestions relate to making their figures slightly more clear, as follows:

Fig. 2 - A key is needed for the different reef areas depicted in the map. What do the "P" on the map refer to? What do the "!" in green on the map refer to?

Fig. 4 - I recommend removing the lines connecting panels a and b. I do not think they are particularly helpful.

Fig. 5a - Define abbreviation in the legend - perhaps refer to Fig. 3 where complete wording on the legend is provided.

Fig. 5b - Define and/or standardize the abbreviations used on the ternary diagram. For example what is "fat" - perhaps for simplicity, use first three letters of each genera? I do not believe there would be an duplicates.

Fig. 6a - slightly shift the legend for "numbers" so it does not overlap with the graph

FIg. 6b - Define abbreviation in the legend - perhaps refer to Fig. 3 where complete wording on the legend is provided.

Fig. 7b, d-g - Perhaps show on the same y-axis scale for easier comparison?

Fig. 7c - Can the authors make this panel larger? It is difficult to read and is a bit blurry.

Additional comments

I have the following suggested edits to the overall text:

Line 67: change "through have a" to "because of its"

Line 80: change "followed" to "following"

Line 97: change "from" to "on"

Lines 112-113: change to read - "...with 25-m transects (Table 1). The smallest sampling was a 7-m belt transect and 2 m2 quadrats..."

Line 114: change to read - "... samples of 11 small and 86 large corals were recorded within a 25-m..."

Lines 119 and 121: delete open and close parentheses "()"

Line 137: correct type to read "monitoring"

Line 140: Change to read "...and second, the effect..."

Line 143: I'm not sure what was meant by "analyses were trialled"?

Line 156: Add hypen to "Thirty-four"

Line 182: Delete typo after "following"

Line 197: Delete "to"

·

Basic reporting

Overall the authors did an excellent job of writing clearly and reporting the work that they conducted. I thought the figures were very clear and easy to interpret.

Experimental design

The design of these experiments are good. The authors fully explained their design and the context of their results.

Validity of the findings

These results are interesting. While bleaching at Mayotte in 2010 has already been documented, I think that the addition of the size class data is novel and an important aspect of bleaching that is under reported. I also hesitate to predict mortality, however the authors have done a good job of putting that analysis into context and provided adequate explanation of the methods of their estimation.

Additional comments

Overall I found this to be an interesting paper. The surveys were conducted well and were done extensively for this island. The bleaching event was put into appropriate context both across time at Moyotte, and in the context of the recent global bleaching events.

Here are a few specific comments of typos or unclear writing for the authors:
Abstract
Line 35-37. This sentence is unclear. Rewrite for clarity.
Introduction
Line 43. No need for a comma here
Line 55. Insert “of the” into this sentence. It should read “…, with most of the world regions predicted..”
Line 67. Replace “through having” with “since it has”
Results
Line 182. There is an extra f at the end of following
Discussion
Line 241. When you say “This paper”, it is not clear if you mean your manuscript or the paper that you describe just before this sentence. Edit for clarity.
Line 350. Delete “the”, you have two of them in this sentence
Conclusions
Line 373. Remove “and” from this sentence.

---

## Round 0.2 · accepted · Accept

I am satisfied with the modifications made to the manuscript and the justifications for any changes suggested by the reviewers that were not made.

#